# Hydrophobicity of Polyacrylate Emulsion Film Enhanced by Introduction of Nano-SiO_2_ and Fluorine

**DOI:** 10.3390/polym11020255

**Published:** 2019-02-03

**Authors:** Tao Xu, Qiangqiang Xiao, Jiayu Chen, Li Li, Xiongjun Yang, Lifang Liu, Wenhui Yuan, Bingjian Zhang, Huijun Wu

**Affiliations:** 1Academy of Building Energy Efficiency, School of Civil Engineering, Guangzhou University, Guangzhou 510006, China; beexu@gzhu.edu.cn (T.X.); 15626426611@163.com (L.L.); wuhuijun@tsinghua.org.cn (H.W.); 2Key Laboratory of Enhanced Heat Transfer and Energy Conservation, The Ministry of Education, School of Chemistry and Chemical Engineering, South China University of Technology, Guangzhou 510640, China; qiang201@163.com (Q.X.); 17051075130@163.com (X.Y.); 3Department of Architecture and Civil Engineering, City University of Hong Kong, Kowloon, Hong Kong; jiaychen@cityu.edu.hk; 4School of Environment Energy, South China University of Technology, Guangzhou 510006, China; lli@scut.edu.cn; 5School of Chemical Engineering and Technology, Sun Yat-sen University, Guangzhou 510275, China; zhbingj@mail.sysu.edu.cn

**Keywords:** polyacrylate emulsion, nano-SiO_2_, fluorine, ultraviolet shielding effect, hydrophobicity

## Abstract

This study proposes to utilize modified Nano-SiO_2_/fluorinated polyacrylate emulsion that was synthesized with a semi-continuous starved seed emulsion polymerization to improve the hydrophobicity, thermal stability, and UV-Vis absorption of polyacrylate emulsion film. To verify the proposed method, a series inspection had been conducted to investigate the features of the emulsion film. The morphological analysis indicated that Nano-SiO_2_ was surrounded by a silane molecule after modification, which can efficiently prevent silica nanoparticles from aggregating. Fourier transform infrared spectra confirmed that modified SiO_2_ and dodecafluoroheptyl methacrylate (DFMA) were successfully introduced to the copolymer latex. The particle size of latex increased with the introduction of modified Nano-SiO_2_ and DFMA. UV-Vis absorption spectra revealed that modified silicon nanoparticles can improve the ultraviolet shielding effect obviously. X-ray photoelectron spectroscopy illustrated that the film–air interface was richer in fluorine than film section and the glass side. The contact angle of modified Nano-SiO_2_/fluorinated polyacrylate emulsion containing 3 wt % DFMA was 112°, slightly lower than double that of polyacrylate emulsion, indicating composite emulsion films possess better hydrophobicity. These results suggest that introducing modified Nano-SiO_2_ and fluorine into polyacrylate emulsion can significantly enhance the thermal stability of emulsion films.

## 1. Introduction

Being widely used in aircraft, functional smart coating, building, navigation, textile, and many other fields, hydrophobic material has attracted considerable attention in recent years [1,2,3]. Due to excellent film appearance, high gloss, good mechanical properties, and ideal adhesive properties with a wide variety of substance, acrylate polymer latex is the most popular coating material [4,5,6]. However, acrylate polymer latex is depreciated when waterproof performance is highly emphasized, since it usually has low hydrophobicity [7]. Therefore, great efforts have been made to modify the structure of polyacrylate emulsion coating to improve its hydrophobicity. Many studies have shown that combining the advantages of inorganic and organic materials to form organic/inorganic composite coatings is an effective method [8,9,10]. Among all inorganic materials, such as SiO_2_, TiO_2_, Al_2_O_3_, ZrO_2_, and CaCO_3_, Nano-SiO_2_ is considered the most promising composite [11]. It can not only significantly improve the mechanical property, heat-resistance, and weather-resistance of a polymer, but also reduce the cost of organic materials and emissions of volatile organic compounds. In addition, owing to the low polarizability and strong electro-negativity of fluorine atoms, fluorine-containing polymer exhibits various merits, such as unique surface and optical properties, high thermal, chemical, and weather resistance [2,12,13]. Therefore, it has been widely applied to material coating [14,15,16,17] and many studies on hydrophobic coatings have focused on silica- and fluorine-modified acrylic polymers [18].

In general, there are two common methods to synthesize Nano-SiO_2_/fluorinated polyacrylate emulsion [19]. For the first method, Liao et al. [20] synthesized silica/polyacrylate hybrid latexes with high silicon contents by directly mixing colloidal silica with polyacrylate emulsion modified by a silane coupling agent. However, heterogeneous dispersion of the inorganic nanoparticles in the polymer matrix is challenging [8,21,22,23]. In addition, the interaction between inorganic nanoparticles and the organic components was weak in such cases. Another method is based on emulsion polymerization in the presence of Nano-SiO_2_. For example, Xu et al. [24] reported a poly (methacrylic-methacrylate)/silica hybrid material with high transparency and heat-stability via a sol-gel process. Chen et al. [25] prepared polyester-based polyurethane/silica composite and showed that the composite obtained by in situ polymerization had better mechanical properties than those achieved by the blending method when silica content is high. Zhou et al. [7] synthesized organic fluorine and Nano-SiO_2_ modified polyacrylate emulsifier-free latex film, via emulsifier-free emulsion polymerization. Although latex film presented excellent mechanical properties and hydrophobicity, current synthesis of Nano-SiO_2_/fluorinated polyacrylate hydrophobic emulsion still has challenges, such as the usage of organic solvent, which can cause environmental problems. The separation and purification process is also time-consuming and can cause secondary reunion of the Nano-SiO_2_, which can result in performance deterioration. Therefore, this study intends to improve the hydrophobicity of polyacrylate emulsion film and simultaneously decrease the amount of organic solvent used in the preparation process, coupling reagent modified Nano-SiO_2_/fluorinated polyacrylate latex particles were synthesized by a semi-continuous starved seed emulsion polymerization with TEOS as precursors for Nano-SiO_2_. The thermal stability and ultraviolet shielding effect of composite and their impact on the hydrophobicity of the latex film were investigated.

## 2. Material Preparation and Experiment Setup

### 2.1. Materials

For the raw materials of this study, the methyl methacrylate (MMA), butyl acrylate (BA), 2-Hydroxyethyl acrylate (HEA) and ethyl silicate (TEOS) of analytical grade were purchased from Shanghai Macklin Biochemical Co. Ltd. (Shanghai, China) and used as the main monomers. The precursors, such as SiO_2_, methacryloyl-propyl trimethoxysilane (KH-570), alkylphenol ethoxylates (OP-10), and dodecafluoroheptyl methacrylate (DFMA) were provided by Aladdin Chemistry Co. Ltd. (Shanghai, China). Potassium persulfate (K_2_S_2_O_8_, analytical grade) was selected as an initiator, while ammonia and ethanol were obtained from Shanghai Resin Co. Ltd. (Shanghai, China). The deionized water (H_2_O) was self-prepared in the laboratory. All agents were used without any further purification.

### 2.2. Preparation and Modification of Silica Nanoparticles

The silica nanoparticles were prepared through the minor modification [26]. With this method, ammonia and ethanol solution were mixed in a three-neck flask in a water bath. The temperature was kept below 50 °C, a certain amount of TEOS was dropwise added to the above solution and stirred for 2 h to acquire silica sol. After several centrifugations and washing with anhydrous ethanol, the silica nanoparticles were successfully prepared. Having been redistributed in the KH-570 solution (5 vol.% KH-570, 45 vol.% ethanol, 50 vol.% H_2_O) and vigorously agitated for 15 min, the Nano-SiO_2_ would be successfully modified by silane. Finally, the modified Nano-SiO_2_ was obtained by high-speed centrifugation followed by vacuum drying at 50 °C for 6 h.

### 2.3. Preparation of Modified Nano-SiO_2_/Fluorinated Polyacrylate Emulsion

The modified Nano-SiO_2_/fluorinated polyacrylate emulsion (for convenience, the modified Nano-SiO_2_/fluorinated polyacrylate emulsion was abbreviated to Si/F polyacrylate emulsion) was successfully synthesized by a semi-continuous starved seed emulsion polymerization. Firstly, the mixture of MMA, BA, HEA, and modified Nano-SiO_2_ was introduced into OP-10 aqueous solution under the condition of vigorous stirring at the 40 °C to form a pre-emulsion I. Then 1/3 pre-emulsion I, H_2_O, and K_2_S_2_O_8_ were added to four-neck flask equipped with a reflux condenser, thermometer, and mechanical stirrer, the reaction was kept at 70 °C for 30 min. Subsequently, the remaining 2/3 pre-emulsion I and K_2_S_2_O_8_ solution were dropwise added to the above mixture within 2 h. After that, the reaction continued for another 1 h. Later, a certain amount of mixture containing MMA, BA, HEA, and DFMA were dropped into the flask within 3 h. Lastly, ammonia was slowly appended into the synthesized emulsion to adjust the pH value to between 6 and 8 to yield stable Si/F polyacrylate emulsion after 2 h continuous reaction at 85 °C. The recipes of polyacrylate emulsion are summarized in Table 1 and Table 2.

### 2.4. Preparation of Latex Film

The Si/F polyacrylate emulsion was cast onto the clean glass plate and formed a uniform latex film at room temperature for 24 h. As shown in the above tables, the reference experiment sample for the modified Nano-SiO_2_/polyacrylate latex film is S5 and for Si/F polyacrylate latex film is F3. In the following discussions, unless otherwise stated, modified Nano-SiO_2_/polyacrylate latex refers to S5 and Si/F polyacrylate latex refers to F3.

### 2.5. Characterization Experiments 

With the experiment samples, several characterization experiments were conducted. The Fourier transform infrared spectra (FT-IR) test was taken on EQUINOX 55 instruments. The morphologies of Nano-SiO_2_, modified Nano-SiO_2_, and composite latex particles were observed, using a transmission electron microscopy (TEM, JEM-2010HR, JEOL, Tokyo, Japan). The contents and elements of the samples were examined with an X-ray photoelectron spectroscopy (XPS, Axis UltraDLD, Shimadzu, Kratos, Japan). The latex particle size distribution was recorded by a ZetaPlus apparatus by Dynamic Light Scattering (DLS, Brookhaven Instrument, NY, USA). The UV-line 9400 type ultraviolet spectrometer was used to test the absorption property of latex films to ultraviolet light. Thermal gravimetric tests (TG 209 F3, NETZSCH, SELB, Germany) were carried out under a nitrogen atmosphere from 25 to 650 °C at a heating rate of 10 °C/min. Contact angles were measured using a sessile drop method on a Dataphysics OCA20CA system (Dataphysics, Filderstadt, Germany) at room temperature. The experiment used deionized water as the probe liquid and the results were acquired 30 s after the dropping of a water drop (3–5 μL) on the latex films.

## 3. Results and Discussion

### 3.1. Structure and Morphology of Nano-SiO_2_, Modified Nano-SiO_2_, and Si/F Polyacrylate Latex

Figure 1a shows the TEM image of Nano-SiO_2_. It can be observed that the Nano-SiO_2_ was densely packed into a large area and exhibits severe aggregations. After modification by KH-570, as shown in Figure 1b,c, the modified silica nanoparticles were presented as separated particles with a diameter of about 50 nm. This can be explained by the silane molecules that on the surface of Nano-SiO_2_ can prevent the aggregation of the particles due to the steric repulsion and a decrease in surface energy of silica nanoparticles [27]. Therefore, it can be concluded that modification using a silane coupling agent can help Nano-SiO_2_ disperse in the polyacrylate composite emulsion. Figure 1d shows that the diameter of Si/F polyacrylate latexes are around 80 nm with core-shell structure, where the modified Nano-SiO_2_ forms the core (black regions) and fluorinated polyacrylate forms the shell (grey regions).

### 3.2. FT-IR Analysis

Figure 2 shows the FT-IR spectra of Nano-SiO_2_, modified Nano-SiO_2_, and Si/F polyacrylate latex. The 1100 cm^−1^ and 468 cm^−1^ peaks in the Nano-SiO_2_ spectrum are the stretching and bending vibration absorption peaks of Si–O–Si, respectively [26]. After modification using KH-570, new characteristic absorption peaks appeared. The peaks located at 2953 cm^−1^ and 2842 cm^−1^ are the stretching vibration of C-H belonging to –CH_3_ and –CH_2_–. The characteristic stretching vibration peak of C=O also appears at 1724 cm^−1^, indicating that KH-570 molecules were successfully grafted on the surface of Nano-SiO_2_ [7,26,27,28]. In Si/F polyacrylate latex spectrum, the characteristic peaks of C–F, Si–O–Si, and ether [(O=C)–O–C] appear in at 1350–1090 cm^−1^, 1250–1000 cm^−1^ and 1075–1020 cm^−1^, respectively. These peaks almost overlapped each other [29]. In addition, the peak emerged at 690 cm^-1^ was attributed to the wagging vibration of C–F bonds [30], indicating that DFMA had taken part in the polymerization reaction to form the Si/F polyacrylate latex. All these analyses illustrate that SiO_2_ and DFMA were successfully introduced into the polyacrylate by semi-continuous starved seed emulsion polymerization.

### 3.3. Particle Size Distribution of Composite Latex

Figure 3 shows the particle size and size distributions of modified Nano-SiO_2_, modified Nano-SiO_2_/polyacrylate latex, and Si/F polyacrylate latex. The average diameter of the modified Nano-SiO_2_ is 52 nm and the modified Nano-SiO_2_/polyacrylate latex and Si/F polyacrylate latex were increased to 76 and 83 nm, respectively. It can be found from Figure 3b,c that there was no modified Nano-SiO_2_/polyacrylate latex or Si/F polyacrylate latex particle that was less than 52 nm. This means that all modified Nano-SiO_2_ particles were surrounded by an organic layer. Figure 4 shows the influence of DFMA contents on the Si/F polyacrylate latex size. When there is no DFMA (0 wt %), the diameter is equal to 76 nm, in accordance with Figure 3b. The diameter of the emulsion particles increased to 86.5 nm as the DFMA amount increased from 0 to 7 wt %, which confirms that Si/F polyacrylate latex presented a core-shell structure where polyacrylate contained fluorine in the shell layer.

### 3.4. Surface Analysis of Composite Polyacrylate Emulsion Films

The surface element composition significantly impacts the surface property and hydrophobicity [31] and Figure 5 shows the XPS spectra of Si/F polyacrylate emulsion film at different interfaces. Figure 5a shows the overall XPS spectrum of the Si/F polyacrylate emulsion film. The peaks at 102.34 eV, 153.47 eV, 284.89 eV, 532.16 eV, and 688.65 eV are corresponding to Si2p, Si2s, C1s, O1s, and F1s, respectively [32]. Figure 5b shows the high-resolution XPS spectrum. The peaks at 293 eV, 288 eV, 286 eV, and 284 eV are corresponding to –CF_3_, –CF, C–O, and C–C [15], respectively. The appearance of silicon (Si2p, Si2s) and fluorine (F1s) signals in Figure 5a indicates that both the SiO_2_ and fluorine element are presented at the surface of the composite films [33]. In addition, it can be observed from Figure 5c that the signal of fluorine in the film–air interface is more intense than that in the film–glass interface, suggesting that the fluorine content in the film–air interface is higher than that in the film–glass interface. Due to low surface energy and the self-aggregated property of the fluorine atom, the fluorinate-contained group preferentially oriented to the film surface during the film formation to decrease surface energy of films [34,35].

### 3.5. Effects of Modified Nano-SiO_2_ on UV Shielding Effect of Composite Emulsion Film

Figure 6 shows the UV-Vis absorption spectra of modified Nano-SiO_2_, fluorinated polyacrylate emulsion film, and Si/F polyacrylate emulsion film. The Si/F polyacrylate emulsion film had a strong absorption capacity in the UV range from 200 to 300 nm, while the absorption capacity is much weaker for fluorinated polyacrylate emulsion without SiO_2_. The broad absorption bands of Si/F polyacrylate emulsion film results from the electron transition within the Nano-SiO_2_, as revealed in the modified Nano-SiO_2_ spectrum [36]. It can also be seen that adding modified Nano-SiO_2_ into polyacrylate emulsion has little impact on the absorption intensity in the visible light region. Thus, it can be concluded that introducing silica nanoparticles into acrylate polymer emulsion can significantly improve the ultraviolet shielding effect of hydrophobic polyacrylate emulsion films.

### 3.6. Thermal Stability of Si/F Polyacrylate Emulsion Films

Thermal weight loss curves of the polyacrylate emulsion film, modified Nano-SiO_2_/polyacrylate emulsion film and Si/F polyacrylate emulsion film are exhibited in Figure 7. It can be seen that the 5% weight reduction of polyacrylate emulsion film, modified Nano-SiO_2_/polyacrylate emulsion film, and Si/F polyacrylate emulsion film happened at 355 °C, 376 °C, and 390 °C, respectively. This indicates that the thermal stability of the polyacrylate emulsion film was increased by introducing SiO_2_ and fluorinated groups. Modified Nano-SiO_2_ can increase the heat resistance of polyacrylate emulsion because of its low thermal conductivity.

In addition, the presence of silicon network (–Si–O–Si–) in the latex films delayed the degradation of composite films, so a higher temperature is required for the same weight loss amount for the modified Nano-SiO_2_/polyacrylate emulsion films and the Si/F polyacrylate emulsion films [37]. When the fluorinated group was introduced to the polyacrylate emulsion, the C–F with high bond energy, concentrating on the surface of the latex particles, can prevent other bonds inside the latex particles from fracturing. Therefore, the thermal stability of polyacrylate emulsion film can be improved by introducing modified Nano-SiO_2_ and fluorinated groups.

### 3.7. Hydrophobicity of Si/F Polyacrylate Emulsion Films

The contact angles can directly reflect the hydrophobic property of a composite. Figure 8 and Table 1 and Table 2 show the effects of DFMA and modified Nano-SiO_2_ contents on contact angles of latex films. It can be observed that the contact angles increased with the increasing modified Nano-SiO_2_ dosages. The increase in modified Nano-SiO_2_ content may result in an increase in the surface roughness of film, and therefore give rise to increasing contact angles [34,38]. When the content of modified Nano-silica exceeded 5 wt %, the contact angle of the hybrid film gradually decreased with the continuing increase of SiO_2_. The possible reason is that modified Nano-SiO_2_ particles were exposed to the surface of the hybrid film, and a portion of Nano-SiO_2_ particles have not been completely modified by KH-570 carried hydroxyl groups. Therefore, 5 wt % SiO_2_ was selected for further research in terms of the effects of DFMA contents on the hydrophobic property. Contact angles of latex films also increased considerably with the increase in fluorine monomer dosages. Due to the low surface energy of fluorine element, side chain fluorine-containing groups preferentially migrated to the surface of latex films in the film forming process [39]. When the DFMA dosage is 3 wt %, the contact angle is 112°. However, when the DFMA exceeds 3 wt %, the increment of contact angles start to shrink because the migration of fluorine atoms onto the film surface is hindered by the steric effect of the fluorine-contained groups [40].

## 4. Conclusions

In this study, hydrophobic Si/F polyacrylate emulsion was successfully synthesized via a semi-continuous starved seed emulsion polymerization. The analysis of the characteristics of the samples indicates that Nano-SiO_2_ particles with an average size of 52 nm have been successfully prepared with the sol-gel method. After the modification process, the modified silicon nanoparticles were evenly dispersed in the polyacrylate emulsion and had intensive interaction with the polyacrylate. Fluorine elements were detected at the surface of the composite polyacrylate emulsion film and the content in the film–air interface was higher than that in the film–glass interface, suggesting the fluorine preferentially concentrated on the film–air interface. The modified Nano-SiO_2_ can provide polyacrylate emulsion film with an obvious UV shielding effect. The thermal stability of polyacrylate emulsion films was also improved by the modified Nano-SiO_2_ and fluorine (5% weight reduction of polyacrylate emulsion film occurred at 355 °C, while Si/F polyacrylate emulsion film at 390 °C). Most importantly, the introduction of modified Nano-SiO_2_ and fluorinate groups can significantly enhance the hydrophobicity of latex films. When the content of modified Nano-SiO_2_ is 5 wt %, the latex film can reach its optimal hydrophobicity. Based on this silica dosage, as the weight fraction of DFMA is increased to 3 wt % from 0, the contact angle will also increase by 22°. The results of this study reveal that the prepared polyacrylate emulsion has great promise in the area of waterproof coating, surface coatings for paper, leather, textiles, and the walls of buildings.

## Figures and Tables

**Figure 1 polymers-11-00255-f001:**
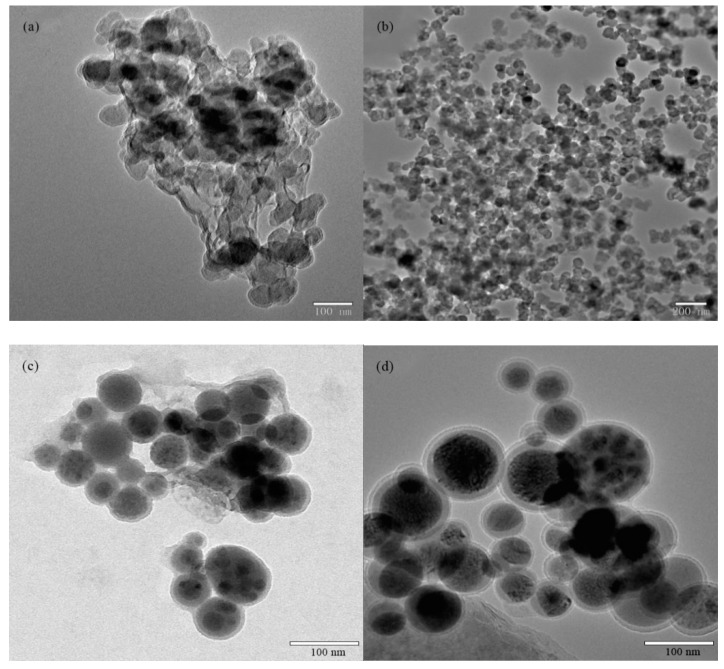
TEM micrographs of (**a**) Nano-SiO_2_, (**b**,**c**) modified Nano-SiO_2_, and (**d**) Si/F polyacrylate latex.

**Figure 2 polymers-11-00255-f002:**
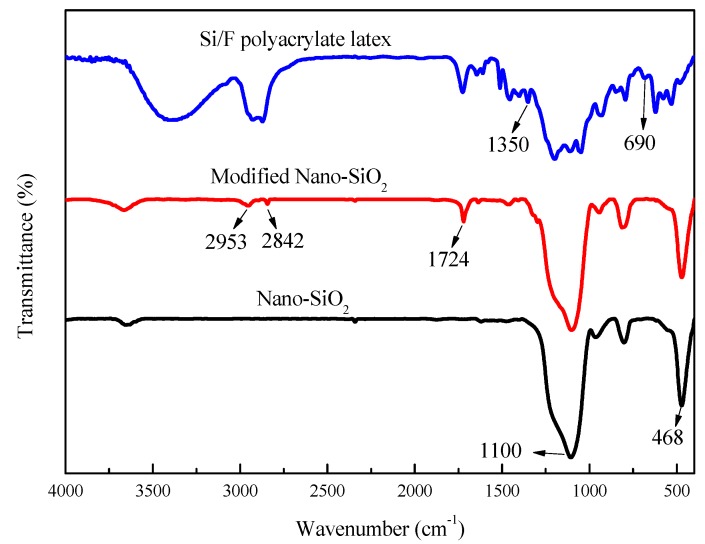
FT-IR spectra of Nano-SiO_2_, modified Nano-SiO_2_, and Si/F polyacrylate latex.

**Figure 3 polymers-11-00255-f003:**
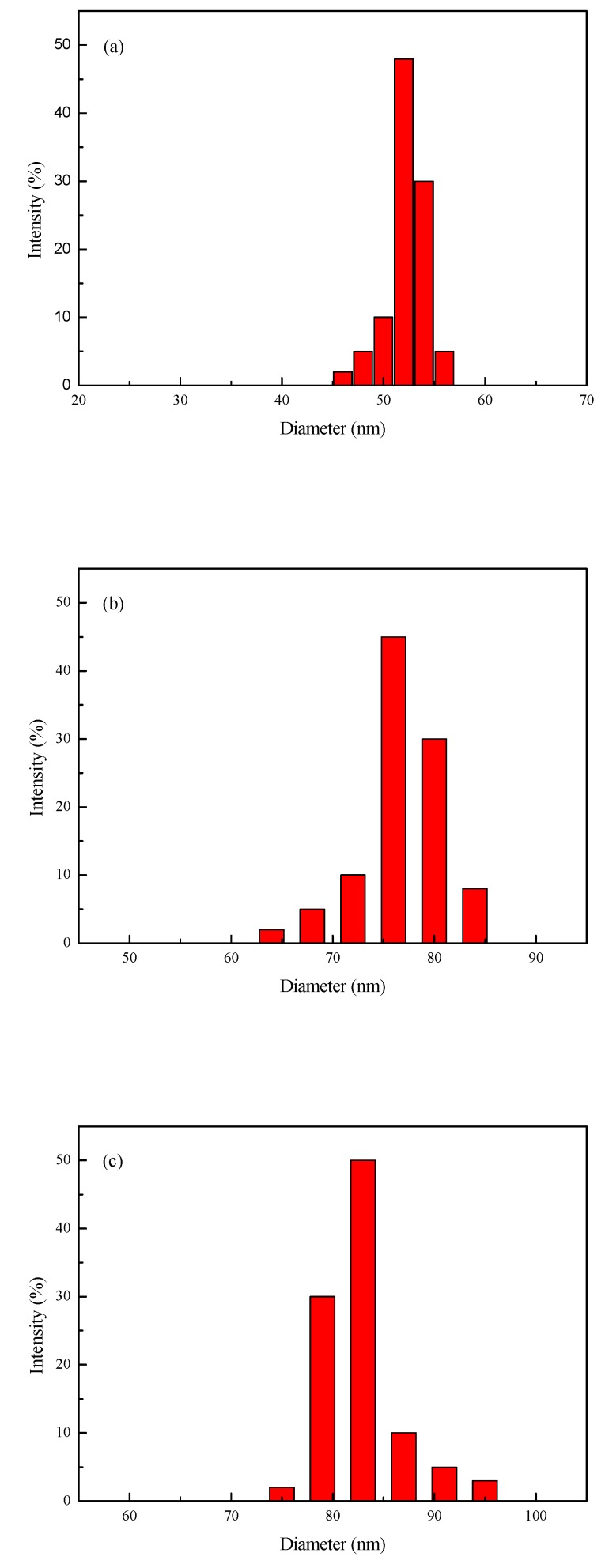
Particle size and distributions of (**a**) modified Nano-SiO_2_, (**b**) modified Nano-SiO_2_/polyacrylate latex, and (**c**) Si/F polyacrylate latex.

**Figure 4 polymers-11-00255-f004:**
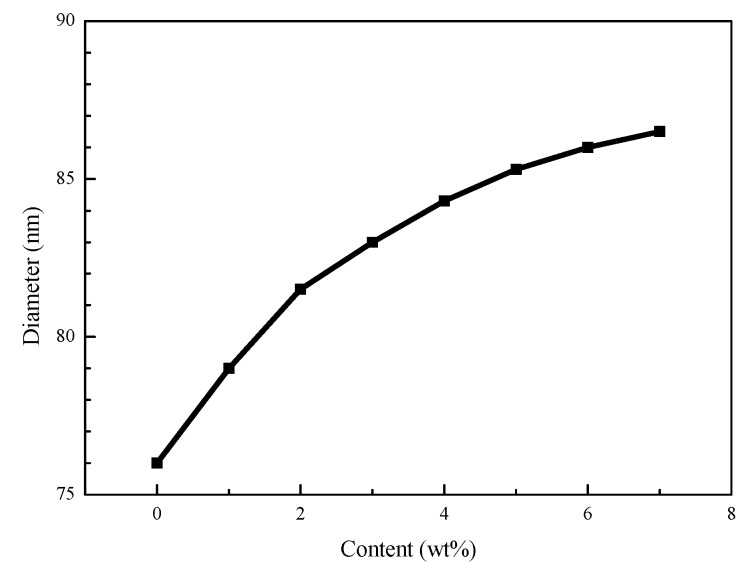
Effects of dodecafluoroheptyl methacrylate (DFMA) dosages on Si/F polyacrylate latex particle size.

**Figure 5 polymers-11-00255-f005:**
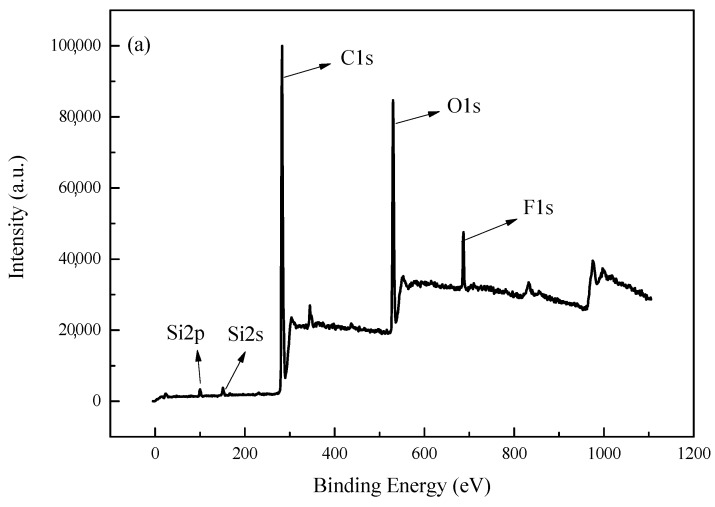
XPS spectra of Si/F polyacrylate emulsion film: (**a**) the films–glass interface, (**b**) High-resolution of C1s signal, and (**c**) F1s signal for film–air interface and film–glass interface.

**Figure 6 polymers-11-00255-f006:**
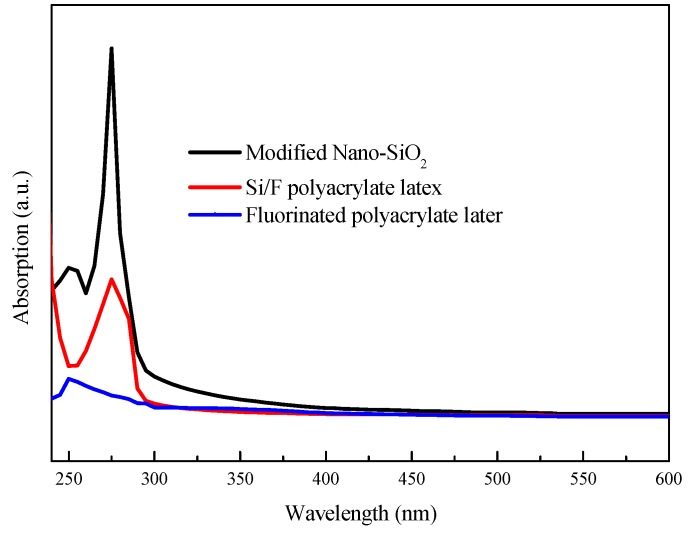
The UV-Vis absorption spectra of modified Nano-SiO_2_, fluorinated polyacrylate emulsion film, and Si/F polyacrylate emulsion film.

**Figure 7 polymers-11-00255-f007:**
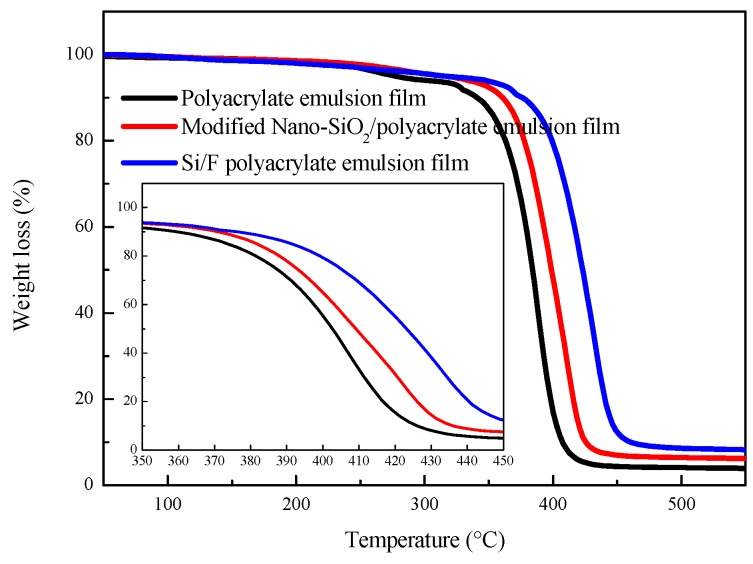
Thermal gravimetric curves of polyacrylate emulsion film, modified Nano-SiO_2_/polyacrylate emulsion film, and Si/F polyacrylate emulsion film.

**Figure 8 polymers-11-00255-f008:**
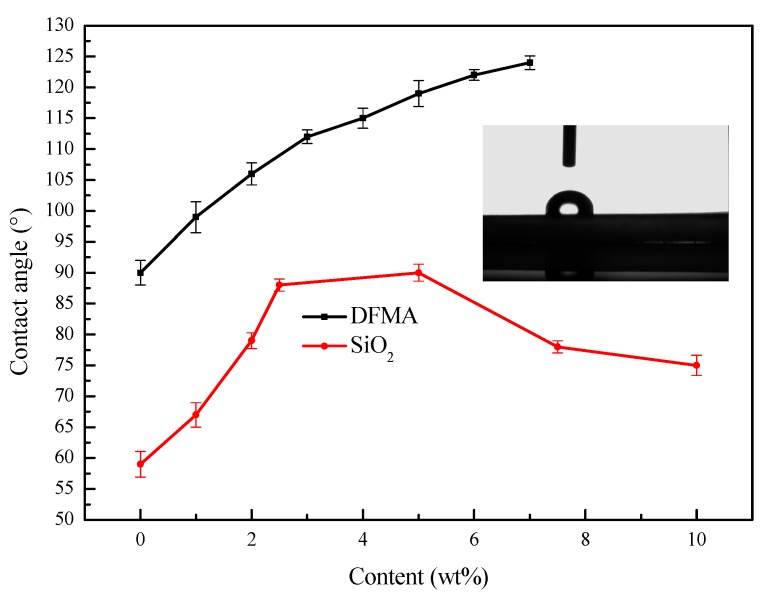
Effects of DFMA and modified Nano-SiO_2_ dosages on contact angles of latex films.

**Table 1 polymers-11-00255-t001:** Recipes for modified Nano-SiO_2_/polyacrylate emulsion without fluorine.

Ingredients	S1	S2	S3	S4	S5	S6	S7
MMA (g)	5	5	5	5	5	5	5
BA (g)	4.8	4.8	4.8	4.8	4.8	4.8	4.8
HEA (g)	1.4	1.4	1.4	1.4	1.4	1.4	1.4
OP-10 (g)	0.8	0.8	0.8	0.8	0.8	0.8	0.8
DFMA (g)	0	0	0	0	0	0	0
H_2_O (g)	100	100	100	100	100	100	100
SiO_2_ (g)	0	1.13	2.29	2.87	5.89	9.08	12.44
SiO_2_ (wt %)	0	1	2	2.5	5	7.5	10
Contact angle (θ)	59	67	79	88	90	78	75

**Table 2 polymers-11-00255-t002:** Recipes for Nano-SiO_2_/fluorinated (Si/F) polyacrylate emulsion.

Ingredients	F1	F2	F3	F4	F5	F6	F7
MMA (g)	5	5	5	5	5	5	5
BA (g)	4.8	4.8	4.8	4.8	4.8	4.8	4.8
HEA (g)	1.4	1.4	1.4	1.4	1.4	1.4	1.4
OP-10 (g)	0.8	0.8	0.8	0.8	0.8	0.8	0.8
DFMA (g)	1.19	2.41	3.65	4.91	6.20	7.52	8.87
H_2_O (g)	100	100	100	100	100	100	100
SiO_2_ (g)	5.89	5.89	5.89	5.89	5.89	5.89	5.89
DFMA (wt %)	1	2	3	4	5	6	7
Contact angle (θ)	99	106	112	115	119	122	124

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
