# Peer review of "Hydrophobicity of Polyacrylate Emulsion Film Enhanced by Introduction of Nano-SiO2 and Fluorine"

_polymers, 2019, doi:10.3390/polym11020255_

Round 1
Reviewer 1 Report
The manuscript entitled “Hydrophobicity of Polyacrylate Emulsion Film Enhanced by Introduction of Nano-SiO2 and Fluorine” is well-written and presents quite good and novel information. However, one of the most the most important question to be improved is the justification in the Introduction topic. Another important question is: If the title claims for hydrophobic aspect, the authors need to improve their scientific data of contact angle and discussion about this fact. The manuscript needs major revision. Following are some of the comments/suggestions to make the paper meaningful.
Comments/Suggestions:
Line 54: You cited: However, heterogeneous dispersion of the inorganic nanoparticles in the polymer matrix is challenging [21]. This reference was published in 2008, eleven years ago. Do you think this heterogeneous dispersion remain as a challenge?
Line 67: In my opinion, you need to improve your justification to show us what is the main novelty of your work compared to the previously published articles.
Line 108: Do you have any information about the quantity of emulsion used for formation of each film by casting? Or any information about the thickness of the films.
Line 113, Characterization experiments: Please, you should explain better some experiments. For example: what are the parameters adopted in FTIR? And for TEM? And for XPS? You need to inform many important characteristics of each experiment for reproduction by other readers/researchers interested in your work.
Line 157: You may combine figure 3 and 4 because they have the same objective, to explain particles size and distribution.
Line 175: You may make the same for XPS images.
Line 208: Where is the derivate curve in the thermogravimetric analysis?
Line 215: You take care about the statement: “This indicates that the thermal stability of polyacrylate emulsion film was increased by introducing SiO2 and fluorinated groups.” Can be this higher thermal stability due to a shielding effect?
Line 231: You cited: The increase in modified Nano-SiO2 contents result in the increase in surface roughness of film, and therefore give rise to increasing contact angles [32,36]. This is true. But, do you think or observe any changes on latex films roughness?
Line 227: Time-resolved contact angle analysis can prove an important characteristic required for hydrophobic surfaces: droplet stability. It means, low absorption/spreading of droplets, showing a real and longer higher/superhydrophobicity. Are you able to show us this behavior in your material?
Author Response
Dear Reviewers,
We would glad to have the opportunity to revise our manuscript and improve the quality of this work. We deeply appreciate all of your time and valuable comments. All comments had been carefully reviewed and the concerns were addressed accordingly. Again, thank you for taking the time and effort to review this manuscript.
Reviewer #1
1. Line 54: You cited: However, heterogeneous dispersion of the inorganic nanoparticles in the polymer matrix is challenging [21]. This reference was published in 2008, eleven years ago. Do you think this heterogeneous dispersion remains as a challenge?
Response: We thank the reviewer’s suggestion. We have reviewed several recent works and the heterogeneous dispersion is still regarded as a major problem can be future properly solved. The updated latest references had been added.
Huang, X.; Liao, W.; Ye, L.; Zhang, N.; Lan, S.; Fan, H.; Qu, J. Fabrication of hydrophobic composite films by sol-gel process between POSS -containing fluorinated polyacrylate latexes and colloidal silica particles. Microporous Mesoporous Mater. 2017, 243, 311–318, doi:10.1016/j.micromeso.2017.02.045.
Qu, A.; Wen, X.; Pi, P.; Cheng, J.; Yang, Z. Synthesis of composite particles through emulsion polymerization based on silica/fluoroacrylate-siloxane using anionic reactive and nonionic surfactants. J. Colloid Interface Sci. 2008, 317, 62–69, doi:10.1016/j.jcis.2007.09.038.
Li, J.; Zhong, S.; Chen, Z.; Yan, X.; Li, W.; Yi, L. Fabrication and properties of polysilsesquioxane-based trilayer core–shell structure latex coatings with fluorinated polyacrylate and silica nanocomposite as the shell layer. J. Coatings Technol. Res. 2018, 15, 1077–1088, doi:10.1007/s11998-018-0044-9.
Zhou, J.; Chen, X.; Duan, H.; Ma, J. Synthesis and characterization of organic fluorine and nano-SiO2 modified polyacrylate emulsifier-free latex. Prog. Org. Coatings 2015, 89, 192–198, doi:10.1016/j.porgcoat.2015.09.016.
2. Line 67: In my opinion, you need to improve your justification to show us what is the main novelty of your work compared to the previously published articles.
Response: We have clarified the statements of the novelty of this work. As the main contribution, this work proposed to prepare polyacrylate emulsion using water as solvent and through the preparation process avoiding the heavy use of organic solvent. By doing so, the polyacrylate emulsion is more eco-friendly.
3. Line 108: Do you have any information about the quantity of emulsion used for formation of each film by casting? Or any information about the thickness of the films.
Response: The thickness of the films was roughly 2-10 um. No special measurement was used to control the thickness of emulsion film and the discussion of the thickness was not focused in this work. Thanks the reviewer’s comments, The investigation and control of thickness precisely is an interesting topic, we will investigate in future.
4. Line 113, Characterization experiments: Please, you should explain better some experiments. For example: what are the parameters adopted in FTIR? And for TEM? And for XPS? You need to inform many important characteristics of each experiment for reproduction by other readers/researchers interested in your work.
Response: We included more details on the characteristics on each experiment, the parameters of FTIR, XEM, XPS were introduced in the revised manuscript.
“With the experiment samples, several characterization experiments were conducted. The fourier transform infrared spectra (FT-IR) test was taken on an EQUINOX 55 instruments in a spectral range 4000-400 cm-1. The morphologies of Nano-SiO2, modified Nano-SiO2, and composite latex particles were observed, using a transmission electron microscopy (TEM, JEM-2010HR) operated at an accelerated voltage of 120 kV. The samples were prepared by drop casting the diluted latex onto 300 mesh Cu grids and drying in air for 5 h. The contents and elements of the samples were examined with an X-ray photoelectron spectroscopy (XPS, Axis UltraDLD) with an Al Kα X-ray source at a take-off angle of 0º. The latex particle size distribution was recorded by a ZetaPlus apparatus by Dynamic Light Scattering (DLS) equipped with a 4mW He-Ne laser. The detector angle is fixed at 90º. The measurement was carried out with 1 mL of diluted latex samples at 25 ºC. The UV-line 9400 type ultraviolet spectrometer was used to test the absorption property of latex films to ultraviolet light. Thermal gravimetric analysis was carried out on a TGA instrument (TG, NETZSCH 209 F3). About 10 mg sample was heated from temperature to 650 ºC at a heating rate of ºC/min under nitrogen atmosphere. Contact angles were measured using a sessile drop method on a Dataphysics OCA20CA system at room temperature. The experiment used deionized water as the probe liquid and the results were acquired 30 s after the dropping of a water drop (3-5 μL) on the latex films.”
5. Line 157: You may combine figure 3 and 4 because they have the same objective, to explain particles size and distribution.
Response: We have merged figure 3 and figure 4 to analyze particles size and distribution, as suggested by the reviewer in Section 3.3.
6. Line 175: You may make the same for XPS images.
Response: We tried to make three images the same in our revised manuscript. However, the coordinate ranges of three XPS images were different with the purpose of making images clear, so these images were not identical to the others.
7. Line 208: Where is the derivate curve in the thermogravimetric analysis?
Response: The thermal stabilities of three films can be determined by TG curves alone, and the weight loss rate of these samples was not discussed, so DTG curve was not given in our manuscript.
8. Line 215: You take care about the statement: “This indicates that the thermal stability of polyacrylate emulsion film was increased by introducing SiO2 and fluorinated groups.” Can be this higher thermal stability due to a shielding effect?
Response: To the best of our knowledge, there is no close relationship between the thermal stability and shielding effect. Also, we cannot find a previous report illustrating that shielding effect can improve the thermal stability.
9. Line 231: You cited: The increase in modified Nano-SiO2 contents result in the increase in surface roughness of film, and therefore give rise to increasing contact angles [32, 36]. This is true. But, do you think or observe any changes on latex films roughness?
Response: Because of the restriction of experimental facilities, we are not able to detect the roughness experimentally. Also, it is true that we cannot observe obvious changes in latex films roughness. In our revised manuscript, we have changed the related sentence to make the explanation more precise.
10. Line 227: Time-resolved contact angle analysis can prove an important characteristic required for hydrophobic surfaces: droplet stability. It means, low absorption/spreading of droplets, showing a real and longer higher/superhydrophobicity. Are you able to show us this behavior in your material?
Response: We thank the reviewer’s critical observation. Yes, time-resolved contact angle analysis is an important characteristic parameter for hydrophobic surfaces, so we are carrying out the research on droplet stability and related results will be published in subsequent papers.

Reviewer 2 Report
Manuscript ID: polymers-421118
Authors: Tao Xu, Qiangqiang Xiao, Jiayu Chen, Li Li, Xiongjun Yang, Lifang Liu, Wenhui Yuan, Bingjian Zhang, Huijun Wu
Title: Hydrophobicity of Polyacrylate Emulsion Film Enhanced by Introduction of Nano-SiO2 and Fluorine
The paper entitled “Hydrophobicity of Polyacrylate Emulsion Film Enhanced by Introduction of Nano-SiO2 and Fluorine” by Wenhui Yuan and his coworkers reported an interesting experimental and modelling discussion in the enhanced high performance of polymer composite by using Nano-SiO2 and fluorine. The finding is significant in manipulating the composites. Although the paper is not a pioneer work in organic fluorine and nano-SiO2 modified polyacrylate emulsion film, it demonstrates the use of simple composite manufacturing processes using inorganic Nano-SiO2 and fluorine to achieve high performance polymer composites.
This manuscript should be revised according to the following comments:
1. There are literatures about the nano-SiO2/fluorinated polyacrylate (e.g., Progress in Organic Coatings 2015, 89, 192-198; Applied Surface Science 2015, 331, 504-511). It will be good to compare the similarity and difference with the related papers.
2. In Fig. 2, the FT-IR spectra should be illustrated in details. Please modify it.
3. The contact angle measurements displayed in the figures are lacking error bars. The authors should indicate the relative reproducibility of their measurements.
4. The applications of such a work ought to be better mentioned.
5. The manuscript appears to have just served the purpose of data collection, but less elaboration or explanation. It will be nicer to have it revised accordingly.
Overall, it is recommended for publication in the Polymers after the major revisions.
Author Response
Dear Reviewers,
We would glad to have the opportunity to revise our manuscript and improve the quality of this work. We deeply appreciate all of your time and valuable comments. All comments had been carefully reviewed and the concerns were addressed accordingly. Again, thank you for taking the time and effort to review this manuscript.
1. There are literatures about the nano-SiO2/fluorinated polyacrylate (e.g., Progress in Organic Coatings 2015, 89, 192-198; Applied Surface Science 2015, 331, 504-511). It will be good to compare the similarity and difference with the related papers.
Response: We would thank the reviewer’s suggestion, we have compared and cited both works in the revised paper. Our work shared the same idea on implementing nano-SiO2/fluorinated polyacrylate, different from previous works, our work focuses on discussing the effect of SiO2 content on the hydrophobicity of polyacrylate emulsion film and the ultraviolet shielding effect caused by Nano-SiO2. Both works have been properly cited.
[7] Zhou, J.; Chen, X.; Duan, H.; Ma, J.; Ma, Y. Synthesis and Characterization of nano-SiO2 modified fluorine-containing polyacrylate emulsifier-free emulsion. Appl. Surf. Sci. 2015, 331, 504–511, doi:10.1016/j.apsusc.2015.01.098.2.
[23] Zhou, J.; Chen, X.; Duan, H.; Ma, J. Synthesis and characterization of organic fluorine and nano-SiO2 modified polyacrylate emulsifier-free latex. Prog. Org. Coatings 2015, 89, 192–198, doi:10.1016/j.porgcoat.2015.09.016.
2. In Fig. 2, the FT-IR spectra should be illustrated in details. Please modify it.
Response: We have modified the Fig. 2 in the revised manuscript. We hope the changes can meet reviewer’s requirement.
3. The contact angle measurements displayed in the figures are lacking error bars. The authors should indicate the relative reproducibility of their measurements.
Response: Thank you for good comment. Error bars were added into the Fig. 8 in revised manuscript.
4. The applications of such a work ought to be better mentioned.
Response: We have mentioned several applications in the first and last sentence of the manuscript.
5. The manuscript appears to have just served the purpose of data collection, but less elaboration or explanation. It will be nicer to have it revised accordingly.
Response: Thank you for this excellent advice. In the revised manuscript, more detailed explanations related to measurement results were given according to your suggestion.

Round 2
Reviewer 1 Report
The authors have modified the manuscript. It can be published in Polymers Journal.
Reviewer 2 Report
The manuscript was revised carefully and improved so much according to reviewers’ suggestions. The scientific insights are expressed well in this manuscript. Overall, the current revision is recommended for publication in the Polymers.